# Comparative Gene Expression Profiles in Parathyroid Adenoma and Normal Parathyroid Tissue

**DOI:** 10.3390/jcm8030297

**Published:** 2019-03-02

**Authors:** Young Jun Chai, Heejoon Chae, Kwangsoo Kim, Heonyi Lee, Seongmin Choi, Kyu Eun Lee, Sang Wan Kim

**Affiliations:** 1Department of Surgery, Seoul Metropolitan Government-Seoul National University Boramae Medical Center, Seoul 07061, Korea; kevinjoon@naver.com; 2Division of Computer Science, Sookmyung Women’s University, Seoul 04310, Korea; heechae@sookmyung.ac.kr (H.C.); sophia5848@naver.com (H.L.); 3Division of Clinical Bioinformatics, Biomedical Research Institute, Seoul National University Hospital, Seoul 03080, Korea; kksoo716@gmail.com (K.K.); sm.alex.choi@gmail.com (S.C.); 4Department of Surgery, Seoul National University Hospital & College of Medicine, Seoul 03080, Korea; kyueunlee@snu.ac.kr; 5Department of Internal Medicine, Seoul National University College of Medicine, and Seoul Metropolitan Government-Seoul National University Boramae Medical Center, Seoul 07061, Korea

**Keywords:** parathyroid adenoma, hyperparathyroidism, gene ontology, parathyroid hormone, parathyroid glands, gene expression profiling, endoplasmic reticulum, RNA, messenger

## Abstract

Parathyroid adenoma is the main cause of primary hyperparathyroidism, which is characterized by enlarged parathyroid glands and excessive parathyroid hormone secretion. Here, we performed transcriptome analysis, comparing parathyroid adenomas with normal parathyroid gland tissue. RNA extracted from ten parathyroid adenoma and five normal parathyroid samples was sequenced, and differentially expressed genes (DEGs) were identified using strict cut-off criteria. Gene Ontology (GO) and Kyoto Encyclopedia of Genes and Genomes (KEGG) pathway enrichment analyses were performed using DEGs as the input, and protein-protein interaction (PPI) networks were constructed using Search Tool for the Retrieval of Interacting Genes/Proteins (STRING) and visualized in Cytoscape. Among DEGs identified in parathyroid adenomas (*n* = 247; 45 up-regulated, 202 down-regulated), the top five GO terms for up-regulated genes were nucleoplasm, nucleus, transcription DNA-template, regulation of mRNA processing, and nucleic acid binding, while those for down-regulated genes were extracellular exosome, membrane endoplasmic reticulum (ER), membrane, ER, and melanosome. KEGG enrichment analysis revealed significant enrichment of five pathways: protein processing in ER, protein export, RNA transport, glycosylphosphatidylinositol-anchor biosynthesis, and pyrimidine metabolism. Further, PPI network analysis identified a densely connected sub-module, comprising eight hub molecules: *SPCS2*, *RPL23*, *RPL26*, *RPN1*, *SEC11C*, *SEC11A*, *RPS25*, and *SEC61G*. These findings may be helpful in further analysis of the mechanisms underlying parathyroid adenoma development.

## 1. Introduction

Primary hyperparathyroidism (PHPT) is a common endocrine disease characterized by inappropriate excessive secretion of parathyroid hormone (*PTH*) and consequent hypercalcemia [1,2]. The prevalence of PHPT is estimated to be 21.1–65.6 per 100,000 person-years [3,4,5,6]. PHPT is more common in women than men, particularly those ≥ 45 years old; the ratio of incidence in women to men is almost 2:1 [3], and PHPT incidence increases rapidly with age. Long-standing PHPT causes serious complications, including urolithiasis, fracture, deterioration of renal function, neuropsychiatric complications, and gastrointestinal symptoms, including constipation and peptic ulcer [7]. Since the introduction of routine serum calcium measurement, health check-ups can identify individuals with completely asymptomatic PHPT.

The most common cause of PHPT is sporadic parathyroid adenoma, which accounts for 85% of PHPT. Parathyroid adenoma is a benign monoclonal tumor, usually involving a single parathyroid gland. Besides sporadic parathyroid adenomas, various hereditary types of hyperparathyroidism are caused by mutations of specific genes, including: multiple endocrine neoplasia type 1 and familial isolated hyperparathyroidism, caused by alterations in the *MEN1* gene; familial hypercalcemia with hypercalciuria, attributable to *CASR* gene changes; and hyperparathyroidism-jaw tumor syndrome and familial isolated hyperparathyroidism, caused by *HRP2* gene mutations [8,9,10].

To date, *MEN1* and *CCND1* (respectively, a tumor suppressor and a proto-oncogene) are the most solidly established molecular drivers of sporadic parathyroid adenoma. Loss of heterozygosity at the *MEN1* locus on chromosome 11q13 is the most common genetic aberration in parathyroid adenoma, occurring in 26.2–50.0% of cases [11,12,13,14]. In addition to *MEN1* and *CCND1*, other genes that participate in the development of parathyroid adenoma include those encoding cyclin-dependent kinase inhibitors, along with *CTNNB1*, *EZH2*, *ZFX*, *GCM2*, and *CASR* [15]. Moreover, recent whole-exome sequencing studies of parathyroid adenoma samples revealed mutations in *POT1* and *RAP1B* [11,16]. Additional genes important for parathyroid tumorigenesis are likely to be identified by next-generation sequence analysis; however, data from transcriptome analysis, to compare gene expression in parathyroid adenomas with that in healthy parathyroid glands, are lacking.

In this study, we performed transcriptome analysis to compare parathyroid adenomas and normal parathyroid glands, with the aim of identifying differentially expressed genes (DEGs). We subsequently used the Database for Annotation, Visualization and Integrated Discovery (DAVID), the Kyoto Encyclopedia of Genes and Genomes (KEGG), and Gene Ontology (GO) databases to systematically extract meaningful biological information from the DEGs identified. We also constructed a protein-protein interaction (PPI) network to ascertain functional relationships between proteins in parathyroid adenoma.

## 2. Materials and Methods

### 2.1. Sample Acquisition

All patients with PHPT had a single parathyroid adenoma, and PHPT was diagnosed based on serum calcium and intact *PTH* levels, in the absence of other possible causes of hypercalcemia. Ultrasound and Sestamibi scans were performed routinely in all patients to localize the tumors and surgical pathology was used to determine whether or not they were parathyroid adenomas. All the PHPT patients were female and the mean age was 53 ± 4.8. The mean levels of serum calcium and intact *PTH* were 12.7 ± 1.0 mg/dL and 460.4 ± 492.9 mmol/L, respectively. Serum magnesium level was not routinely checked. The mean size of a parathyroid adenoma was 2.3 ± 1.1 cm. None of the patients had evidence of familial disease, a history of neck irradiation, renal insufficiency, or cardiovascular disorders. Fresh frozen parathyroid adenoma tissue was obtained during parathyroidectomy at Seoul National University Hospital, between July 2014 and September 2016.

Normal parathyroid glands were obtained from thyroid carcinoma patients who underwent thyroidectomy at Seoul National University-Seoul Metropolitan Government (SNU-SMG) Boramae Medical Center, between July 2016 and January 2017. All the patients who underwent thyroidectomy were female and the mean age was 50 ± 7.4. The mean levels of serum calcium and intact *PTH* were 8.7 ± 0.5 mg/dL and 34.7 ± 17.0 mmol/L, respectively. The normal parathyroid tissues were collected after intraoperative frozen biopsy for microscopic tissue confirmation, due to macroscopic ambiguity, as previously described in the literature [17]. Tissue samples were snap frozen in liquid nitrogen and stored at −80 °C. This study was approved by the institutional review board of SNU-SMG Boramae Medical Center (L-2018-311).

### 2.2. RNA Sequencing

Total RNA samples were extracted using an RNeasy mini kit (Qiagen, Hilden, Germany), according to the manufacturer’s recommendations. RNA quality was assessed by analysis of rRNA band integrity, using an Agilent RNA 6000 Nano kit (Agilent Technologies, Santa Clara, CA, USA). RNA libraries were constructed using a TruSeq RNA Access Library Prep kit (Illumina, San Diego, CA, USA), according to the manufacturer’s protocol. The size and quality of libraries were evaluated by electrophoresis using an Agilent High Sensitivity DNA kit (Agilent Technologies); fragments were 350–450 bp. Subsequently, libraries were sequenced using an Illumina HiSeq2500 sequencer (Illumina). Sequencing data were deposited in NCBI’s Sequence Read Archive (SRA) and are accessible through BioProject accession number PRJNA516535.

### 2.3. Identification of DEGs

Quality control of FASTQ data was performed using FastQC [18]. Paired-end reads (101 bp) were aligned to the human genome (GRCh37/hg19 assembly) using STAR (v. 2.4.1d, https://github.com/alexdobin/STAR) mapper [19]. Gene expression levels were then calculated as fragments per kilobase per million reads (FPKMs) using cufflinks [20], with LNCipedia and refFlat databases for region annotations [21,22]. All FPKM values were assigned a pseudocount of 1, then transformed to the log2 scale. Gene data were filtered out if median log transformed expression was <3 in both tumor and control samples. DEGs were then defined as genes with fold-change ≥2 and false discovery rate-adjusted *p* values < 0.001.

### 2.4. Functional and KEGG Pathway Enrichment Analysis

Functional analyses were performed using DAVID (http://david.abcc.ncifcrf.gov/), a functional annotation tool that allows investigators to unravel the biological meaning behind an input gene list [23]. Based on the extracted DEGs, GO associations were analyzed in three categories: biological process (BP), cellular component (CC), and molecular function (MF) [24]. The KEGG pathway database was used to identify biological pathways enriched for the identified DEGs. Statistical significance was evaluated using the Fisher exact test [25], and *p* < 0.05 was regarded as significant.

### 2.5. Protein-Protein Interaction (PPI) Network Construction and Module Analysis

To ascertain functional interactions between proteins in parathyroid cells and extract functional modules, PPI networks were constructed using the Search Tool for the Retrieval of Interacting Genes/Proteins (STRING) database, based on identified DEGs (medium confidence score < 0.5) [26]. For further pruning and identification of a core module from the constructed PPI network, the MCODE algorithm, a recursive vertex weighting scoring scheme, that measures local network density, assigning higher scores to nodes whose immediate neighbors are more interconnected, was used [27]. Default pruning thresholds (degree cut-off = 2, node score cut-off = 0.2, k-core = 2, and max depth = 100) were implemented to identify a core module. A detailed scoring formula had been described previously [27]. Finally, the constructed PPI network and identified core modules were visualized using Cytoscape, a tool for network data integration and visualization [28].

## 3. Results

### 3.1. Characteristics of Study Subjects and Mutation Screening

Ten parathyroid adenomas and five normal parathyroid tissue samples were analyzed. The characteristics of the study subjects are presented in Table 1. Somatic mutations in genes associated with hyperparathyroidism, including *MEN1*, *CASR*, *AP2S1*, *GNA11*, *CDC73*, *CDKN1A*, *CDKN1B*, *CDKN2C*, *RET*, *PTH*, *CCND1*, *AIP*, *CTNNB1*, *EZH2*, *ZFX*, *CDC73*, and *FGF23*, were not detected in either parathyroid adenoma or normal parathyroid tissues.

### 3.2. Genes Differentially Expressed between PHPT and Normal Parathyroid Tissue

The mRNA expression values for genes evaluated in the ten parathyroid adenoma and five normal parathyroid tissue samples are shown in Appendix A. A total of 247 DEGs (45 up-regulated and 202 down-regulated) were identified in parathyroid adenomas (Appendix A). The top DEGs in parathyroid adenoma, compared with normal parathyroid gland, are presented in Table 2.

In addition, we investigated the expression levels of selected genes that have been implicated in parathyroid function (Table 3). Levels of Klotho (*KL*) and *PTH* mRNA were significantly lower in parathyroid adenoma than in normal parathyroid tissue, while those of *CASR*, *FGFR1*, *FGFR2*, and *VDR* did not differ significantly.

### 3.3. Gene Ontology (GO) Functional and KEGG Pathway Enrichment Analysis

Using the 45 genes up-regulated in parathyroid adenoma as input for analysis using DAVID, 16 GO terms that satisfied the cut-off criteria (*p* < 0.05) were identified (Table 4); six, two, and eight GO terms were from the BP, CC, and MF ontology categories, respectively. Nucleoplasm, nucleus, transcription DNA-template, regulation of mRNA processing, and nucleic acid binding were the most significantly enriched GO terms among up-regulated genes. Using the 202 genes down-regulated in parathyroid adenoma as input, 78 GO terms were extracted (Appendix A), of which 32, 37, and 9 were from the BP, CC, and MF ontology categories, respectively. The top five most significantly enriched of these terms in each GO category are presented in Table 5. Extracellular exosome, membrane endoplasmic reticulum (ER), membrane, ER, and melanosome were the most significantly enriched GO terms among down-regulated genes.

Five pathways were identified as significantly enriched by KEGG analysis for down-regulated DEGs, including protein processing in ER (15 genes), protein export (5 genes), RNA transport (8 genes), glycosylphosphatidylinositol-anchor biosynthesis (3 genes), and pyrimidine metabolism (5 genes) (Table 6).

### 3.4. Protein-Protein Interaction (PPI) Network Construction and Module Analysis

The PPI network of genes up-regulated in parathyroid adenoma, consisting of 13 nodes and 8 edges (average node degree = 1.231; average local clustering coefficient = 0.00) is illustrated in Appendix A. The PPI network constructed using down-regulated genes, which comprised 120 nodes and 230 edges (average node degree = 3.833; average local clustering coefficient = 0.229) was shown in Appendix A. The most significant PPI sub-module was selected from sub-modules sorted by interaction score using clustering analysis (Figure 1); it included eight hub molecules (*SPCS2*, *RPL23*, *RPL26*, *RPN1*, *SEC11C*, *SEC11A*, *RPS25*, *SEC61G*), which suggests these hub molecules might play crucial roles in development of parathyroid adenoma. Of them, *RPL23* had the highest score, indicating that it was the most strongly interconnected with its neighbor proteins.

## 4. Discussion

To better understand the genetics involved in parathyroid adenoma development, we characterized the global expression profiles in parathyroid adenomas and normal parathyroid tissues by transcriptome analysis. A total of 247 DEGs were identified by comparison of ten sporadic parathyroid adenoma and five normal parathyroid tissue samples. Among those genes up-regulated in parathyroid adenomas, some, such as *MED12*, *KMT5A*, *BMP2K*, and *ATAD2*, are implicated in cell proliferation and transcriptional regulation, supporting the notion that they potentially contribute to tumorigenesis.

The *MED12* protein encoded by *MED12* is a component of the *CDK8* subcomplex, which is essential for *CDK8* kinase activation [29]. In addition, recent studies revealed frequent mutations in *MED12*, exon 2, in both benign and malignant tumors [30,31]. Further, the protein encoded by *KMT5A* is a protein-lysine N-methyltransferase that can monomethylate Lys-20 of histone H4 (*H4K20* methyltransferase) to repress transcription of various genes. This enzyme is an important regulator of the cell cycle [32], and *KMT5A* expression is elevated in different types of cancer tissues and cancer cell lines, including those of bladder cancer, non-small cell and small cell lung carcinoma, chronic myelogenous leukemia, hepatocellular carcinoma, papillary thyroid cancer, and pancreatic cancer [33,34]. A recent study demonstrated that cancer cell growth is significantly suppressed by a reduction or loss of *KMT5A*-mediated methylation of proliferation cell nuclear antigen (PCNA), a widely recognized cell proliferation marker of tumor progression including that of parathyroid adenoma [35], suggesting that *KMT5A*-dependent PCNA methylation might promote the development of parathyroid adenoma. Therefore, *KMT5A* should be further validated as a possible biomarker for parathyroid adenoma progression or development.

The active form of vitamin D, 1,25-dihydroxyvitamin D (1,25(OH)_2_D_3_), is a potent regulator of parathyroid proliferation. A recent study revealed that 1,25(OH)_2_D_3_ treatment induces miR-1228, a *BMP2K* targeting factor, in a dose-dependent manner in human osteoblasts [36]. Though there is still no evidence for the parathyroid gland, vitamin D is presumed to interact with the BMP2 pathway during parathyroid proliferation or growth. In addition, a large body of evidence shows that 1,25(OH)_2_D_3_ alters the cell cycle, ultimately affecting pRB proteins and the *E2F* family of transcription factors [37]. *ATAD2*, an *E2F* target gene, binds to the *MYC* oncogene and stimulates its transcriptional activity [38]. We speculate that *ATAD2* could link E2F and MYC pathways and contribute to parathyroid tumor development.

The *KL* and *PTH* genes, which play significant roles in calcium regulation, were down-regulated in parathyroid adenoma compared with normal parathyroid tissue samples. *KL* encodes the Klotho protein, which is a type I transmembrane protein, and *FGF23* co-receptor. Our results are consistent with previously published data, demonstrating decreased expression of Klotho mRNA and protein in parathyroid adenomas [39,40]. *KL* is strongly expressed in tissues that require abundant calcium transport, including distal convoluted tubule cells in kidneys, choroid plexus, and parathyroid glands [41]. As deletion of *KL* functions in parathyroid gland hyperplasia, this gene may also contribute to the development of parathyroid adenoma [42]. By contrast, *FGF23* a Klotho ligand, was not identified as a DEG in this study.

Interestingly, the *PTH* gene exhibited reduced expression, despite the higher serum *PTH* levels in patients with parathyroid adenomas. This suggests that parathyroid adenoma cells are less efficient at producing *PTH* relative to normal parathyroid cells; hence the higher levels of serum *PTH* in individuals with parathyroid adenoma may simply reflect that these tumors contain more cells that produce *PTH* than normal parathyroid tissue. This is consistent with previous studies, which reported that parathyroid adenomas have weaker *PTH* mRNA intensity relative to normal parathyroid tissue, and that disease severity is dependent on tumor weight [43,44].

Proteins function through interaction with other proteins; thus investigation of PPI networks is essential for understanding biological processes. Hub nodes, identified by computational measurement of their degree of relationship to neighbor nodes, are considered to have pivotal roles in networks. Hub molecules identified in this study were associated with signal peptidase complex subunit (*SPCS2*), ribosomal proteins (*RPL23*, *RPL26*, *RPN1*, *RPS25*), and the ER membrane (*SEC11C*, *SEC11A*, *SEC61G*).

*PTH* is secreted by exocytosis via secretory vesicles. Most secretory proteins are proteolytically processed by signal peptidases, to remove their signal peptides, and proper secretory protein cleavage is critical for the efficient function of many protein-secreting cells [45,46]. Prepro*PTH* is synthesized as a 115-amino acid precursor and converted to pro-*PTH* upon cleavage of its signal peptide within the ER. The results of our PPI analysis suggest the importance of the proteolytic process of signal peptide cleavage in the ER in the development of parathyroid adenoma. Moreover, GO analysis also suggested that down-regulated genes in parathyroid adenoma are implicated in signal peptide processing, based on BP ontology terms. Notably, *PTH* was associated with *KL* and *GJA1* (connexin 43) proteins in PPI analysis.

This study has limitations. First, the analysis included a relatively small number of samples; hence there is potential for the results to have been greatly influenced by each sample or individual patient characteristics. To minimize the effects of sex and age, we selected both parathyroid adenoma and normal parathyroid samples from middle-aged female patients. PHPT is observed predominantly in women, although the reason for this is still unclear. It would be interesting to check the influence of gender using a linear regression-based statistical model in a further study with a larger sample size [47]. Moreover, to rule out the effects of critical genes in the development of parathyroid adenoma, we tested for somatic mutations in all genes reported as associated with hyperparathyroidism. The second limitation of this study was the lack of validation testing of DEGs using an independent test set. Such validation analysis was not possible, as there is no publicly available gene database for parathyroid adenoma and normal parathyroid, and because of budget limitations. We attempted to mitigate this limitation by applying strict cut-off criteria for identification of DEGs.

## 5. Conclusions

We identified DEGs in parathyroid adenoma compared with normal parathyroid tissue and GO terms, which may contribute to the development or progression of parathyroid adenoma. Further, PPI network analysis highlighted a densely connected PPI sub-module, consisting of eight hub molecules. Additional validation of these findings, in a larger number of samples and using functional experiments, is required to confirm the results of this study.

## Figures and Tables

**Figure 1 jcm-08-00297-f001:**
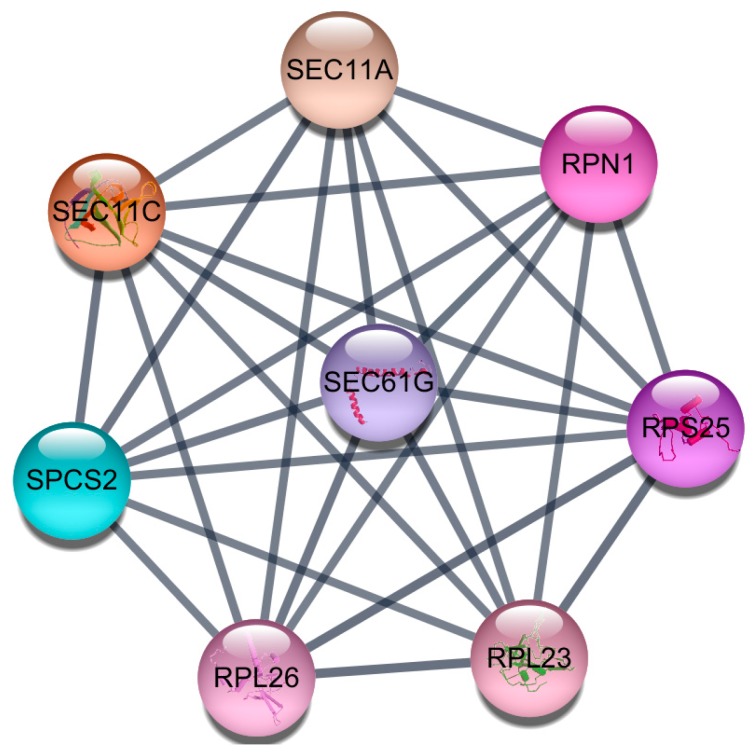
The most significant protein-protein interaction (PPI) sub-module detected by the MCODE algorithm, which includes eight hub molecules, Signal Peptidase Complex Subunit 2 (*SPCS2*), Ribosomal Protein L23 (*RPL23*), Ribosomal Protein L26 (*RPL26*), Ribophorin I (*RPN1*), SEC11 Homolog C, Signal Peptidase Complex Subunit (SEC11C), SEC11 Homolog A, Signal Peptidase Complex Subunit (*SEC11A)*, Ribosomal Protein S25 (*RPS25*), and Sec61 Translocon Gamma Subunit (*SEC61G*). These hub molecules might play crucial roles in development of parathyroid adenoma. Of them, *RPL23* had the highest score, indicating that it was the most strongly interconnected with its neighbor proteins.

**Table 1 jcm-08-00297-t001:** Characteristics of the study subjects.

Group	Gender	Age, Years	Tumor Size, cm	Serum Intact *PTH*, mmol/L	Serum Calcium, mg/dL	Serum Phosphorous, mg/dL	Serum 25(OH)_2_D_3_, ng/mL	Serum 1,25(OH)_2_D_3_, pg/mL
PHPT group	Female	52	2.5	350	12.6	1.9	11.5	57.2
	Female	63	1.5	173	11.4	3.1	6.0	130.1
	Female	55	2.1	262	12.1	2.5	30.5	82.0
	Female	47	2.2	286	12.0	2.4	12.9	53.0
	Female	51	3.5	1596	14.2	2.5	8.7	30.3
	Female	56	3.8	204	12.6	2.3	12.4	69.2
	Female	47	0.9	120	11.3	2.6	21.1	59.5
	Female	56	0.6	213	12.8	2.5	22.9	69.0
	Female	50	2.1	267	13.9	2.9	17.9	69.4
	Female	53	3.8	1133	13.8	1.8	10.0	34.3
Normal group	Female	61		40	9.3	3.9	-	15.6
	Female	50		29	9.0	3.3	20.6	69.9
	Female	46		28	8.5	3.3	-	30.0
	Female	41		61	8.0	4.1	-	51.5
	Female	50		15.7	8.9	3.6	19.1	25.2

PHPT: primary hyperparathyroidism; *PTH*: parathyroid hormone; 25(OH)_2_D_3_: 25-dihydroxyvitamin D_3_; 1,25(OH)_2_D_3_: 1,25-dihydroxyvitamin D_3_.

**Table 2 jcm-08-00297-t002:** Top 20 differentially expressed genes in parathyroid adenoma with respect to the normal parathyroid gland.

Expression	Gene	Fold Change	*p* Value × 10	FDR
Up-regulated	*BMP2K* (BMP2 Inducible Kinase)	5.67	5.90 × 10^−8^	6.39 × 10^−5^
	*MED12* (Mediator Complex Subunit 12)	4.79	5.38 × 10^−8^	6.39 × 10^−5^
	*NUFIP1* (Nuclear FMR1 Interacting Protein 1)	2.47	1.33 × 10^−7^	8.55 × 10^−5^
	*KRBOX4* (KRAB Box Domain Containing 4)	3.60	2.96 × 10^−7^	0.00011
	*ATAD2* (ATPase Family, AAA Domain Containing 2)	3.70	5.41 × 10^−7^	0.000156
	*GPBP1* (GC-Rich Promoter Binding Protein 1)	2.85	6.49 × 10^−7^	0.000158
	*LUC7L* (LUC7 Like)	2.07	1.16 × 10^−6^	0.000176
	*TCHP* (Trichoplein Keratin Filament Binding)	2.76	1.20 × 10^−6^	0.000176
	*GOLGA8Q* (Golgin A8 Family Member Q)	2.80	2.69 × 10^−6^	0.00025
	*CCDC174* (Coiled-Coil Domain Containing 174)	2.02	2.97 × 10^−6^	0.000262
	*ZNF674* (Zinc Finger Protein 674)	3.63	3.09 × 10^−6^	0.000264
	*CTTNBP2* (Cortactin Binding Protein 2)	5.49	4.82 × 10^−6^	0.00035
	*ARIH2* (Ariadne RBR E3 Ubiquitin Protein Ligase 2)	5.14	5.16 × 10^−6^	0.000363
	*GOLGA8O* (Golgin A8 Family Member O)	2.71	6.00 × 10^−6^	0.00039
	*PPM1B* (Protein Phosphatase, Mg2+/Mn2+ Dependent 1B)	4.14	6.18 × 10^−6^	0.000392
	*ZNF605* (Zinc Finger Protein 605)	2.94	6.92 × 10^−6^	0.000401
	*COPS7B* (COP9 Signalosome Subunit 7B)	3.61	8.75 × 10^−6^	0.000451
	*KMT5A* (Lysine Methyltransferase 5A)	3.00	9.42 × 10^−6^	0.000459
	*NPIPA7* (Nuclear Pore Complex Interacting Protein Family Member A7)	2.22	9.86 × 10^−6^	0.000459
	*SLTM* (SAFB Like Transcription Modulator)	2.52	9.34 × 10^−6^	0.000459
Down-regulated	*DEGS1* (Delta 4-Desaturase, Sphingolipid 1)	−6.93	1.95 × 10^−8^	6.39 × 10^−5^
	*TMBIM6* (Transmembrane BAX Inhibitor Motif Containing 6)	−7.28	4.65 × 10^−8^	6.39 × 10^−5^
	*SSBP3* (Single Stranded DNA Binding Protein 3)	−7.37	6.67 × 10^−8^	6.39 × 10^−5^
	*SNORA74A* (Small Nucleolar RNA, H/ACA Box 74A)	−46.50	7.37 × 10^−8^	6.39 × 10^−5^
	*DPYD* (Dihydropyrimidine Dehydrogenase)	−5.02	9.45 × 10^−8^	7.02 × 10^−5^
	*ALG5* (ALG5, Dolichyl-Phosphate Beta-Glucosyltransferase)	−15.16	1.48 × 10^−7^	8.55 × 10^−5^
	*ZNF552* (Zinc Finger Protein 552)	−5.20	1.96 × 10^−7^	0.000102
	*NBEAL1* (Neurobeachin Like 1)	−3.90	2.46 × 10^−7^	0.00011
	*OGN* (Osteoglycin)	−18.30	2.67 × 10^−7^	0.00011
	*CALR* (Calreticulin)	−6.21	2.90. × 10^−7^	0.00011
	*ZNF33A* (Zinc Finger Protein 33A)	−4.17	3.83. × 10^−7^	0.000133
	*SPDYE16* (Speedy/RINGO Cell Cycle Regulator Family Member E16)	−12.79	4.24 × 10^−7^	0.000138
	*FZD6* (Frizzled Class Receptor 6)	−3.90	4.53 × 10^−7^	0.000139
	*ATP6AP2* (ATPase H+ Transporting Accessory Protein 2)	−21.89	5.72 × 10^−7^	0.000157
	*PIGG* (Phosphatidylinositol Glycan Anchor Biosynthesis Class G)	−3.80	6.47 × 10^−7^	0.000158
	*UBAP1* (Ubiquitin Associated Protein 1)	−3.57	6.68 × 10^−7^	0.000158
	*ST13* (ST13, Hsp70 Interacting Protein)	−3.20	7.75 × 10^−7^	0.000175
	*YTHDC2* (YTH Domain Containing 2)	−3.57	8.17 × 10^−7^	0.000176
	*CPE* (Carboxypeptidase E)	−10.34	9.18 × 10^−7^	0.000176
	*PTH* (Parathyroid Hormone)	−6.62	9.36 × 10^−7^	0.000176

FDR: false discovery rate.

**Table 3 jcm-08-00297-t003:** mRNA expression levels of selected genes with parathyroid-related functions in parathyroid adenomas and parathyroid tissues.

Gene	Parathyroid Adenomas	Normal Parathyroid	Fold Change	*p* Value	Adjusted *p* Value
*CASR* (Calcium Sensing Receptor)	174.02 (59.59, 345.05)	394.47 (265.2, 601.57)	0.44	0.004	0.004
*FGFR1* (Fibroblast Growth Factor receptor1)	4.75 (0.10, 30.60)	7.99 (2.66, 9.54)	0.59	0.9	0.25
*FGFR2* (Fibroblast Growth Factor Receptor2)	5.37(1.94, 10.06)	16.02 (10.00, 23.48)	0.34	0.01	0.007
*KL* (Klotho)	3.33 (0.23, 7.76)	55.02 (19.47, 89.49)	0.06	<0.001	<0.001
*PTH* (Parathyroid Hormone)	2216.46 (1083.31, 5942.02)	16,100 (12,266.3, 24,066.2)	0.14	<0.001	<0.001
*VDR* (Vitamin D Receptor)	12.86 (4.23, 44.80)	13.05 (10.38, 36.49)	0.99	0.54	0.168

Values were expressed in median (range).

**Table 4 jcm-08-00297-t004:** Enriched Gene Ontology (GO) terms of the up-regulated genes in parathyroid adenoma.

Category	GO Term ID	GO Term	Count	*p* Value
Biological process ontology	GO:0006351	transcription, DNA-templated	14	<0.001
	GO:0050684	regulation of mRNA processing	3	<0.001
	GO:0006325	chromatin organization	3	0.004
	GO:0045893	positive regulation of transcription, DNA-templated	6	0.005
	GO:0018026	peptidyl-lysine monomethylation	2	0.017
	GO:0006355	regulation of transcription, DNA-templated	8	0.038
Cellular component ontology	GO:0005654	nucleoplasm	19	<0.001
	GO:0005634	nucleus	26	<0.001
Molecular function ontology	GO:0003676	nucleic acid binding	10	<0.001
	GO:0003682	chromatin binding	6	0.002
	GO:0003713	transcription coactivator activity	5	0.002
	GO:0000166	nucleotide binding	5	0.008
	GO:0003677	DNA binding	10	0.012
	GO:0003690	double-stranded DNA binding	3	0.015
	GO:0003729	mRNA binding	3	0.034
	GO:0016279	protein-lysine N-methyltransferase activity	2	0.039

**Table 5 jcm-08-00297-t005:** Top 5 most significantly enriched GO terms of the down-regulated genes in parathyroid adenoma according to GO term categories.

Category	GO Term ID	GO Term	Count	*p* Value
Biological process ontology	GO:0002474	antigen processing and presentation of peptide antigen via MHC class I	5	<0.001
	GO:0061077	chaperone-mediated protein folding	5	<0.001
	GO:0006465	signal peptide processing	4	0.002
	GO:0006506	GPI anchor biosynthetic process	4	0.003
	GO:0044829	positive regulation by host of viral genome replication	3	0.003
Cellular component ontology	GO:0070062	extracellular exosome	79	<0.001
	GO:0016020	membrane	64	<0.001
	GO:0005789	endoplasmic reticulum membrane	36	<0.001
	GO:0005783	endoplasmic reticulum	31	<0.001
	GO:0042470	melanosome	12	<0.001
Molecular function ontology	GO:0044822	poly(A) RNA binding	23	0.003
	GO:0005515	protein binding	110	0.004
	GO:0001540	beta-amyloid binding	4	0.005
	GO:0019904	protein domain specific binding	8	0.006
	GO:0051087	chaperone binding	5	0.01

**Table 6 jcm-08-00297-t006:** Enriched Kyoto Encyclopedia of Genes and Genomes (KEGG) pathways of the down-regulated genes in parathyroid adenomas.

Term ID	Term	Count	*p* Value	Genes
hsa04141	Protein processing in endoplasmic reticulum	15	<0.001	*SEC63*, *DNAJA1*, *EIF2AK1*, *XBP1*, *UGGT2*, *CANX*, *SEC61G*, *STT3A*, *PDIA6*, *HYOU1*, *SEC13*, *RPN1*, *PDIA3*, *CALR*, *BCAP31*
hsa03060	Protein export	5	<0.001	*SEC63*, *SPCS2*, *SEC61G*, *SEC11A*, *SEC11C*
hsa03013	RNA transport	8	0.008	*STRAP*, *XPO1*, *EIF3E*, *SUMO1*, *EIF2S3*, *NUP205*, *SEC13*, *RGPD2*
hsa00563	Glycosylphosphatidylinositol (GPI)-anchor biosynthesis	3	0.043	*PIGU*, *PIGG*, *PIGP*
hsa00240	Pyrimidine metabolism	5	0.049	*CTPS2*, *NT5C3A*, *NME7*, *POLR2H*, *DPYD*

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
