# Peer review of "Comparative Gene Expression Profiles in Parathyroid Adenoma and Normal Parathyroid Tissue"

_jcm, 2019, doi:10.3390/jcm8030297_

Round 1
Reviewer 1 Report
A control group, i.e. muscle tissue, has to be included.
Scientific writing is poor when using abbreviations.
Scientific writing is poor when using figures that do not transport any message.
Line 59: please replace neoplasia with adenoma. DO the authors want to deal with parathyroid cancer?
Line 60: the comment on treatment is not credible. In clinical practice it is extremely difficult to localize early changes of the parathyroid glands. How could an intervention take place?
Section 2.1 is quite incomplete, i.e. insufficient. Please describe the patients with more detail: age, sex, diagnostic procedures. Did the study include sonography or maybe MIBI scan or maybe choline PET? What was the size of the adenomas? What were the laboratory findings? Were there any problems in recognizing the normal parathyroids in the cases of thyroid carcinoma? How were the borders of the thyroid tumors in relation to the parathyroids? Again basic demographic data is missing: age, sex, PTH levels, calcium levels, diagnostic procedures. Please include the blood levels of magnesium also. One cannot jump to Table 1 in order to find some information.
Table 1: abbreviations should appear in the text. The incomplete addition of a – lost – subtitle is insufficient.
Table 2: nobody can gain any kind of information from the abbreviated results. Please add a description of the gene to the table. Again take more care in writing the manuscript and define an abbreviation in the text. What do the authors mean by FDR?????
Table 3: nobody can get a clear idea of the results when looking at a string of numbers. The Table has to be re-written by someone with experience in scientific writing. Why did the authors choose to use 9 decimals for the p-values? Does it make any intelligent difference? The same applies to the Fold change values.
Why did the authors choose the term ontology for the Tables? A common definition of the term is:
Ontology is the philosophical field revolving around (the study of) the nature of reality (all that is or exists), and the different entities and categories within reality.
Please explain whether the authors are approaching the topic from a philosophical point of view.
What does Figure 1 try to demonstrate? The balls are just hanging around without any explanation of the abbreviations which by the way are not easily read. I cannot get any information from Figure 2 either. Is this matrix necessary to explain that only 3 balls fit into a red box? Make it more scientific!
The same fault regarding abbreviations is found in Figure 3.
There is no independent control tissue in the study, i.e. muscle.
Author Response
Reviewer 1:
Point 1: Line 59: please replace neoplasia with adenoma. DO the authors want to deal with parathyroid cancer?
Response 1: We replaced neoplasia with adenoma as the Reviewer recommended in line 60.
Point 2: Line 60: the comment on treatment is not credible. In clinical practice it is extremely difficult to localize early changes of the parathyroid glands. How could an intervention take place?
Response 2: We share the Reviewer's concern. We have edited the text about treatment as follows.
Identification of specific genes involved in parathyroid tumorigenesis is important, because a sophisticated understanding of the molecular mechanisms that accelerate parathyroid adenoma development may ultimately provide insights that improve diagnosis (lines 59-61).
Point 3: Section 2.1 is quite incomplete, i.e. insufficient. Please describe the patients with more detail: age, sex, diagnostic procedures. Did the study include sonography or maybe MIBI scan or maybe choline PET? What was the size of the adenomas? What were the laboratory findings?
Response 3: We have provided more details including age, sex, use of sonography, MIBI scan, the sizes of the adenomas, and laboratory findings to section 2.1 Sample acquisition is described below. Choline PET was not used.
Line 83-87: Ultrasound and MIBI scans were performed routinely in all patients to localize the tumors and surgical pathology was used to determine whether or not they were parathyroid adenomas. All the PHPT patients were female and the mean age was 53 ± 4.8. The mean levels of serum calcium and intact PTH were 12.7 ± 1.0 mg/dL and 460.4 ± 492.9 mmol/L, respectively. The mean size of a parathyroid adenoma was 2.3 ± 1.1 cm.
Points 4: Were there any problems in recognizing the normal parathyroids in the cases of thyroid carcinoma? How were the borders of the thyroid tumors in relation to the parathyroids? Again basic demographic data is missing: age, sex, PTH levels, calcium levels, diagnostic procedures. Please include the blood levels of magnesium also. One cannot jump to Table 1 in order to find some information.
Response 4: Differentiating normal parathyroid tissue from lymph nodes during thyroid surgery is challenging even for an experienced endocrine surgeon because their colors and shapes are similar. Therefore, during thyroid surgery, surgeons cut a very small part of the tissue for frozen section to determine whether the tissue is the parathyroid gland or lymph node. This procedure is common and important especially during thyroid cancer surgery because metastatic lymph nodes should not be mistaken for parathyroid glands and should be removed.
The normal parathyroid tissues obtained in this study were distant from the thyroid carcinoma and tumor invasion was not suspected. This is why we sent the tissues for frozen section and could preserve the tissue in the operative field when it was confirmed as parathyroid tissue. Therefore, the normal parathyroid tissues used in this study can be considered as ‘real’ normal parathyroid unaffected by thyroid carcinoma.
In fact, obtaining normal parathyroid tissues in this way is a common procedure among endocrine researchers (reference: Carling et al. J Clin Endocrinol Metab. 2000;85:2000-2003). We have added a reference in line 97. Our institute (SMG - SNU Boramae Medical Center) has more than 100 thyroid cancer cases per year, and thus we have plenty of small pieces of frozen normal parathyroid tissues that were sent for frozen sectioning and storage in the Department of Pathology.
Unfortunately, we do not check serum magnesium levels routinely. Therefore, we cannot provide this information.
We have added more details including age, sex, and laboratory findings of the patients with normal parathyroid tissues as follows.
Line 93-95: All the patients who undertook thyroidectomy were female and the mean age was 50 ± 7.4. The mean levels of serum calcium and intact PTH were 8.7 ± 0.5 mg/dL and 34.7 ± 17.0 mmol/L, respectively.
Point 5: Table 1: abbreviations should appear in the text. The incomplete addition of a – lost – subtitle is insufficient.
Response 5: “PHPT” and “PTH”, which appear as abbreviations in Table 1, were written out in full in the first sentence of the Introduction. However, we have re-designed Table 1 according to the Reviewer’s suggestion, and written out all abbreviations in the legend at the bottom of Table 1. We also corrected the erroneous expression of “25(OH)2D3” unit.
Point 6: Table 2: nobody can gain any kind of information from the abbreviated results. Please add a description of the gene to the table. Again take more care in writing the manuscript and define an abbreviation in the text. What do the authors mean by FDR?????
Response 6: As the Reviewer suggested, we have written out the full names of genes and what FDR means at the bottom of Table 2.
Point 7: Table 3: nobody can get a clear idea of the results when looking at a string of numbers. The Table has to be re-written by someone with experience in scientific writing. Why did the authors choose to use 9 decimals for the p-values? Does it make any intelligent difference? The same applies to the Fold change values.
Response 7: We thank the Reviewer for these helpful comments. The genes in the Table 3 are well-known to be involved in parathyroid hormone and calcium homeostasis. Therefore, authors believe that showing the expression levels of these genes is valuable and informative to readers. We redesigned Table 3 according to the suggestions of the Reviewer, and we now show P values to 3 decimals, fold-change values, and expression levels to 2 decimals.
Point 8: Why did the authors choose the term ontology for the Tables? A common definition of the term is: Ontology is the philosophical field revolving around (the study of) the nature of reality (all that is or exists), and the different entities and categories within reality. Please explain whether the authors are approaching the topic from a philosophical point of view. What does Figure 1 try to demonstrate? The balls are just hanging around without any explanation of the abbreviations which by the way are not easily read. I cannot get any information from Figure 2 either. Is this matrix necessary to explain that only 3 balls fit into a red box? Make it more scientific!
Response 8: The meaning of ‘Ontology’ in ‘Gene ontology’ is different from the meaning of ontology used in philosophy. Gene ontology is a term used in bioinformatics, which is a major bioinformatics initiative to unify the representation of gene and gene product attributes across all species (reference: the Gene Ontology project in 2008. Nucleic Acids Res. 2008;36:D440-444. PMCID:PMC2238979). Please check references describing how ‘gene ontology’ is used in bioinformatics (Zhou et al. Identification of molecular target genes and key pathways in hepatocellular carcinoma by bioinformatics analysis. Onco Targets Ther. 2018;11:1861-9; Dai et al. Gene expression profiles and pathway enrichment analysis of human osteosarcoma cells exposed to sorafenib. FEBS Open Bio. 2018;8(5):860-7).
We redesigned Figure 1, 2, and we have increased the font size of the gene names to make them more readable in Figure 1 & 2. We have also have added explanations of the PPI network to Figure 1 & 2, and added three nodes to the red box in Figure 2, as the Reviewer suggested.
Point 9: The same fault regarding abbreviations is found in Figure 3.
Response 9: We have now written out the full names of genes in Figure 3.
Point 10: There is no independent control tissue in the study, i.e. muscle.
Response 10: In studies comparing normal samples of a certain tissue and tumorous samples of the tissue, an independent control tissue is not necessary. This is because the purpose of the study was to examine the mRNA expression of all genes and identify those that were differentially expressed between normal and tumorous samples. In this study, our aim was to investigate the mRNA expressions levels of genes involved in calcium and parathyroid hormone regulation, such as CASR, FGFR, KL, PTH, and VDR in normal parathyroid tissue and parathyroid adenoma tissue. We did not include an independent control because the gene expression of an independent control would have been totally irrelevant, and would not have been informative.
Reviewer 2 Report
Review of „Comparative gene expression profiles in parathyroid adenoma and normal parathyroid tissue” by Chai.
Based on the transcriptome analyses performed on 5 normal and 10 parathyroid adenoma tissues, the authors aimed to identify the relevant differentially expressed genes between both groups and their possible involvement in various biological pathways.
Employment of RNA sequencing, gene ontology (GO) and Kyoto Encyclopedia of Genes and Genomes (KEGG) pathway, revealed that genes identified in parathyroid adenomas were mostly related with the processes of transcription and regulation of mRNA metabolism, while the genes decreased in PHPT were associated with exosome and melanosome function, and endoplasmatic reticulum. Similar results were obtained with GO and KEGG analyses.
This is an interesting study providing an overview about the differential transcript expression between normal and parathyroid adenoma tissues. However, I have several comments and suggestions:
1. Did the authors include the influence of the gender in the statistical analyses? It would be important, since all patients investigated are women. It would be interesting to check it with a linear regression-based statistical model. Second, since the normal parathyroid tissues were obtained due to the thyroidectomy, there is a possibility that they could be also changed.
2. Did the PHPT patients reveal any signs of renal insufficiency or cardiovascular disorders?
3. I would recommend writing down in the Table 2, additionally to the names of the genes, the full names of the DEGs.
4. Could the authors provide any information concerning in vitro or animal studies related with the DEGs? I mean whether overexpression or knock-out of the specific genes affected the genes identified by the authors.
5. Could the authors state more precisely which of the identified genes should be further validated as possible biomarkers for disease progression and/or development?
Author Response
Reviewer 2:
Point 1: Did the authors include the influence of the gender in the statistical analyses? It would be important, since all patients investigated are women. It would be interesting to check it with a linear regression-based statistical model. Second, since the normal parathyroid tissues were obtained due to the thyroidectomy, there is a possibility that they could be also changed.
Response 1: We thank the Reviewer for these helpful comments. We selected female samples to reduce genetic variation between males and females since the number of samples was relatively small. In addition, primary hyperparathyroidism is observed predominantly in women, although the reason for this is still unclear. I agree that it would be valuable to check the influence of gender using a linear regression-based statistical model in a further study with a large sample size. We have mentioned this point in the Discussion, lines 287-290 as follows.
PHPT is observed predominantly in women, although the reason for this is still unclear. It would be interesting to check the influence of gender using a linear regression-based statistical model in a further study with a large sample size.
We share the Reviewer’s concern about normal parathyroid tissues. To explain why we obtained normal parathyroid tissues during thyroidectomy, it is important to take into consideration that differentiating normal parathyroid tissue from lymph nodes during thyroid surgery is challenging even for an experienced endocrine surgeon because their colors and shapes are similar. Therefore, during thyroid surgery, surgeons cut a very small part of the tissue for frozen section to determine whether the tissue is the parathyroid gland or lymph node. This procedure is common and important especially during thyroid cancer surgery because metastatic lymph nodes should not be mistaken for parathyroid glands and should be removed.
The normal parathyroid tissues obtained in this study were distant from the thyroid carcinoma and tumor invasion was not suspected. This is why we sent the tissues for frozen section and could preserve the tissue in the operative field when it was confirmed as parathyroid tissue. Therefore, the normal parathyroid tissues used in this study can be considered as ‘real’ normal parathyroid unaffected by thyroid carcinoma.
In fact, obtaining normal parathyroid tissues in this way is a common procedure among endocrine researchers (reference: Carling et al. J Clin Endocrinol Metab. 2000;85:2000-2003). We have added a reference in line 97.
Point 2: Did the PHPT patients reveal any signs of renal insufficiency or cardiovascular disorders?
Response 2: There were no signs of renal insufficiency or cardiovascular disorders in the study subjects. We have mentioned this point in “Materials and Methods”, lines 87-88 as follows.
None of the patients had evidence of familial disease, a history of neck irradiation, renal insufficiency, or cardiovascular disorders.
Point 3: I would recommend writing down in the Table 2, additionally to the names of the genes, the full names of the DEGs.
Response 3: We thank the Reviewer for this suggestion and have added the full names of the DEGs to Table 2.
Point 4: Could the authors provide any information concerning in vitro or animal studies related with the DEGs? I mean whether overexpression or knock-out of the specific genes affected the genes identified by the authors.
Response 4: Thank you for this insightful query. We think this is a valid point. Therefore, to support the results of the current study, we searched the literature again for in vitro or animal studies on BMP2K and ATAD2 genes, which were up-regulated in parathyroid adenomas in our study. We have modified the “Discussion” to include this information. Please see lines 245-252 and additional references.
The active form of vitamin D, 1,25‐dihydroxyvitamin D [1,25(OH)2D3], is a potent regulator of parathyroid proliferation. A recent study revealed that 1,25(OH)2D3 treatment induces miR-1228, a BMP2K targeting factor, in a dose-dependent manner in human osteoblasts [36], suggesting that vitamin D might interact with the BMP2 pathway during parathyroid proliferation or growth. In addition, a large body of evidence shows that 1,25(OH)2D3 alters the cell cycle, ultimately affecting pRB proteins and the E2F family of transcription factors [37]. ATAD2, an E2F target gene, binds to the MYC oncogene and stimulates its transcriptional activity [38], suggesting that it could link E2F and MYC pathways and contribute to parathyroid tumor development.
Point 5: Could the authors state more precisely which of the identified genes should be further validated as possible biomarkers for disease progression and/or development?
Response 5: We thank the Reviewer for this suggestion. According to the Reviewer’s suggestion, we now suggest that KMT5A could be used a biomarker for parathyroid adenoma progression or development. We have added this information to the Discussion, lines 236-244 together with relevant references.
KMT5A expression is elevated in different types of cancer tissues and cancer cell lines, including those of bladder cancer, non-small cell and small cell lung carcinoma, chronic myelogenous leukemia, hepatocellular carcinoma, papillary thyroid cancer, and pancreatic cancer [33, 34]. A recent study demonstrated that cancer cell growth is significantly suppressed by a reduction or loss of KMT5A-mediated methylation of PCNA (proliferation cell nuclear antigen), a widely recognized cell proliferation marker of tumor progression including that of parathyroid adenoma [35], suggesting that KMT5A-dependent PCNA methylation might promote the development of parathyroid adenoma. Therefore, KMT5A should be further validated as a possible biomarker for parathyroid adenoma progression or development.
Round 2
Reviewer 1 Report
The authors have not made sufficient changes to the study. It remains speculative and lacks a control.
Lines 59-61: the authors remain by their utopic goals. This is simply unrealistic.
Lines 83-88: there is some improvement however there is no mention of magnesium levels.
Line 93: false wording, the patients cannot undertake an operation.
Figure 1: a convolute of round elements with abbreviations. What does it mean for clinicians?
Figure 2: not one single clinician will ever profit from this artwork.
Figure 3: minimal improved graphic with a little bit of explanation. What does it mean for clinicians?
Lines 245-252: the statements have to be clearly declared as purely speculative. The authors should clearly differentiate between in-vitro models and real-life medicine.
Lines: 287-290: there is no literature support for this speculative proposal.
Author Response
Point 1: Lines 59-61: the authors remain by their utopic goals. This is simply unrealistic.
Response 1: We share the Reviewer’s concern. We deleted the sentences.
Point 2: Lines 83-88: there is some improvement however there is no mention of magnesium levels.
Response 2: We described as follows in the previous ‘response to reviewer’.
“Unfortunately, we do not check serum magnesium levels routinely. Therefore, we cannot provide this information”
We additionally mention on the manuscript line 85-86 as follows.
“Serum magnesium level was not routinely checked”
Point 3: Line 93: false wording, the patients cannot undertake an operation.
Response 3: We have changed ‘undertook’ to ‘underwent’.
Point 4: Figure 1: a convolute of round elements with abbreviations. What does it mean for clinicians?
Response 4: We share the concern and we have deleted the figure and supplied as Supplemental Figure 1.
Point 5: Figure 2: not one single clinician will ever profit from this artwork.
Response 5: According to reviewer’s suggestion, we have deleted the figure and supplied as Supplemental Figure 2.
Point 6: Figure 3: minimal improved graphic with a little bit of explanation. What does it mean for clinicians?
Response 6: We have added the clinical significance of the 8 hub molecules as follows (Line 194-195 and line 208-209)
which suggests these hub molecules might play crucial roles in development of parathyroid adenoma.
Point 7: Lines 245-252: the statements have to be clearly declared as purely speculative. The authors should clearly differentiate between in-vitro models and real-life medicine.
Response 7: We share again the Reviewer’s concern and have modified the statements.
The active form of vitamin D, 1,25‐dihydroxyvitamin D [1,25(OH)2D3], is a potent regulator of parathyroid proliferation. A recent study revealed that 1,25(OH)2D3 treatment induces miR-1228, a BMP2K targeting factor, in a dose-dependent manner in human osteoblasts [36]. Though there is still no evidence for parathyroid gland, vitamin D is presumed to interact with the BMP2 pathway during parathyroid proliferation or growth. In addition, a large body of evidence shows that 1,25(OH)2D3 alters the cell cycle, ultimately affecting pRB proteins and the E2F family of transcription factors [37]. ATAD2, an E2F target gene, binds to the MYC oncogene and stimulates its transcriptional activity [38]. We speculate that ATAD2 could link E2F and MYC pathways and contribute to parathyroid tumor development.
Point 8: Lines: 287-290: there is no literature support for this speculative proposal.
Response 8: According to the Reviewer’s suggestion, we added relevant reference.
PHPT is observed predominantly in women, although the reason for this is still unclear. It would be interesting to check the influence of gender using a linear regression-based statistical model in a further study with a large sample size (Predictors of Short-Term Changes in Serum Intact Parathyroid Hormone Levels in Hemodialysis Patients: Role of Phosphorus, Calcium, and Gender. J Clin Endocrinol Metab. 1998 Nov;83(11):3860-6).
Reviewer 2 Report
I have no further comments.
Author Response
Thank you.